# Epidermal cell turnover across tight junctions based on Kelvin's tetrakaidecahedron cell shape

Mariko Yokouchi[1,2], Toru Atsugi[1,3], Mark van Logtestijn[4], Reiko J Tanaka[4], Mayumi Kajimura[5,6], Makoto Suematsu[5,6], Mikio Furuse[7,8], Masayuki Amagai[1,9]*, Akiharu Kubo[1]*

[1]Department of Dermatology, Keio University School of Medicine, Tokyo, Japan; [2]Nerima General Hospital, Tokyo, Japan; [3]KOSÉ Corporation, Tokyo, Japan; [4]Department of Bioengineering, Faculty of Engineering, Imperial College London, London, United Kingdom; [5]Department of Biochemistry, Keio University School of Medicine, Tokyo, Japan; [6]Suematsu Gas Biology Project, Exploratory Research for Advanced Technology, Japan Science and Technology, Tokyo, Japan; [7]Division of Cell Structure, National Institute for Physiological Sciences, Okazaki, Japan; [8]Department of Physiological Sciences, SOKENDAI (The Graduate University for Advanced Studies), Okazaki, Japan; [9]RIKEN Center for Integrative Medical Sciences, Yokohama, Japan

**Abstract** In multicellular organisms, cells adopt various shapes, from flattened sheets of endothelium to dendritic neurons, that allow the cells to function effectively. Here, we elucidated the unique shape of cells in the cornified stratified epithelia of the mammalian epidermis that allows them to achieve homeostasis of the tight junction (TJ) barrier. Using intimate in vivo 3D imaging, we found that the basic shape of TJ-bearing cells is a flattened Kelvin's tetrakaidecahedron (f-TKD), an optimal shape for filling space. In vivo live imaging further elucidated the dynamic replacement of TJs on the edges of f-TKD cells that enables the TJ-bearing cells to translocate across the TJ barrier. We propose a spatiotemporal orchestration model of f-TKD cell turnover, where in the classic context of 'form follows function', cell shape provides a fundamental basis for the barrier homeostasis and physical strength of cornified stratified epithelia.

*For correspondence: amagai@med.keio.ac.jp (MA); akiharu@keio.jp (AK)

**Competing interests:** The authors declare that no competing interests exist.

## Introduction

The epidermis of the skin is a stratified epithelial cellular sheet that forms physical barriers on the body surface. Epidermal barrier dysfunctions cause not only lethal congenital disorders, but also a predisposition to the development of allergic diseases (*McGrath and Uitto, 2008*; *Akiyama, 2010*; *Kubo et al., 2012*; *Weidinger and Novak, 2015*). The mammalian epidermis has two main physical barriers, an air-liquid interface barrier formed by the stratum corneum (SC) and a liquid-liquid interface barrier formed by tight junctions (TJs) (*Kubo et al., 2012*). The TJ is a specialized intercellular adhesion complex that is crucial for epidermal barrier function, as it seals the paracellular space of epithelial cellular sheets (*Tsukita et al., 2001*; *Van Itallie and Anderson, 2014*; *Krug et al., 2014*). Maintenance of the TJ barrier is crucial for proper formation of the SC (*Sugawara et al., 2013*), and thus for maintaining the physical barriers of the skin.

The epidermis consists of keratinocytes that proliferate only in the basal layer and move upward to sequentially form stratum spinosum, stratum granulosum (SG), and SC, and finally shed off from

**eLife digest** The skin surface – known as the epidermis – is made up of sheets of cells that are stacked up in layers. One of the roles of the skin is to provide a protective barrier that limits what leaks into or out of the body. A particular layer of the epidermis – referred to as the stratum granulosum – is primarily responsible for forming this barrier. The cells in this layer are sealed together in a zipper-like fashion by structures known as tight junctions.

New skin cells are continuously produced in the lowest cell layers of the epidermis, and move upwards to integrate into the stratum granulosum layer to replace old cells (which also move upwards to leave the layer). How stratum granulosum cells are replaced without disrupting the tight junction barrier was not well understood.

Yokouchi et al. used a technique called confocal microscopy to examine the stratum granulosum cells in the ears of mice, and found that the shape of these cells forms the basis of the barrier that they form. These cells resemble a flattened version of a shape called Kelvin's tetrakaidecahedron: a 14-sided solid with six rectangular and eight hexagonal sides. This structure was proposed by Lord Kelvin in 1887 to be the best shape for filling space. Tight junctions are present on the edges of the flattened Kelvin's tetrakaidecahedron.

Further experiments revealed that the tight junctions move from cell to cell in a spatiotemporally-coordinated manner in order to maintain a continuous barrier throughout the stratum granulosum as cells are replaced. A newly formed stratum granulosum cell appears beneath the cell that it will replace. The shape of these cells enables a new barrier of three-way tight junction contacts to form between them and the neighboring cells in the stratum granulosum. After this barrier has formed, the upper cell leaves the stratum granulosum.

Future research could address how cells adopt the flattened Kelvin's tetrakaidecahedron shape, and discover why tight junctions only form in one layer of the epidermis.

the top layer of the SC (*Figure 1A*). During the course of its upward movement, a keratinocyte becomes flattened at the SG, forms TJs with adjacent cells in the SG2 layer (the second cell layer of the SG), loses its TJs in the SG1 layer (the top layer of the SG) and is finally cornified to form the SC (*Figure 1A*) (*Kubo et al., 2012*; *Yoshida et al., 2013*). Despite its critical importance for the skin homeostasis, the TJ barrier of the epidermis consists of only a single layer of TJs forming a honey-comb-like mesh (TJ honeycomb) in the SG2 layer (*Figure 1A*) (*Furuse et al., 2002*; *Kubo et al., 2009*; *Yoshida et al., 2013*). How can the TJ barrier be maintained while keratinocytes translocate across the single-layered TJ honeycomb for cell turnover?

In this study, we uncovered a mechanism of spatiotemporal coordination that replaces TJs from one cell to another to maintain TJ barrier homeostasis during cell turnover in the stratified epithelium of the epidermis. This model provides a fundamental structural basis for the integrity, physical strength, and homeostasis of the epidermis.

## Results

### Double-edged TJ polygons are observed in the single-layered epidermal TJ honeycomb

To elucidate the mechanism of TJ barrier homeostasis in the epidermis, we investigated the three-dimensional (3D) structure of the TJ honeycomb and its time-dependent changes. TJs were visualized in whole-mounted epidermis prepared from mouse-ear skin via immunostaining of zonula occludens-1 (ZO-1), an intracellular TJ scaffold protein (*Figure 1B*). The murine ear epidermis consists of a basal layer, a spinous layer, and three layers of SG cells, and exhibits a regular structure of vertically aligned SC cells (corneocytes) and SG cells (*Mackenzie, 1969*; *1975*; *Kubo et al., 2009*). Two-dimensional (2D)-projected images of the whole-mounted epidermis confirmed previous observations of one single-layered TJ honeycomb in the epidermis (*Figure 1B*).

We noticed that a significant number of TJ polygons in the TJ honeycomb of murine ear epidermis were double-edged (9.8 ± 0.6%, mean ± SEM, five independent assays; *Figure 1B*). The 3D

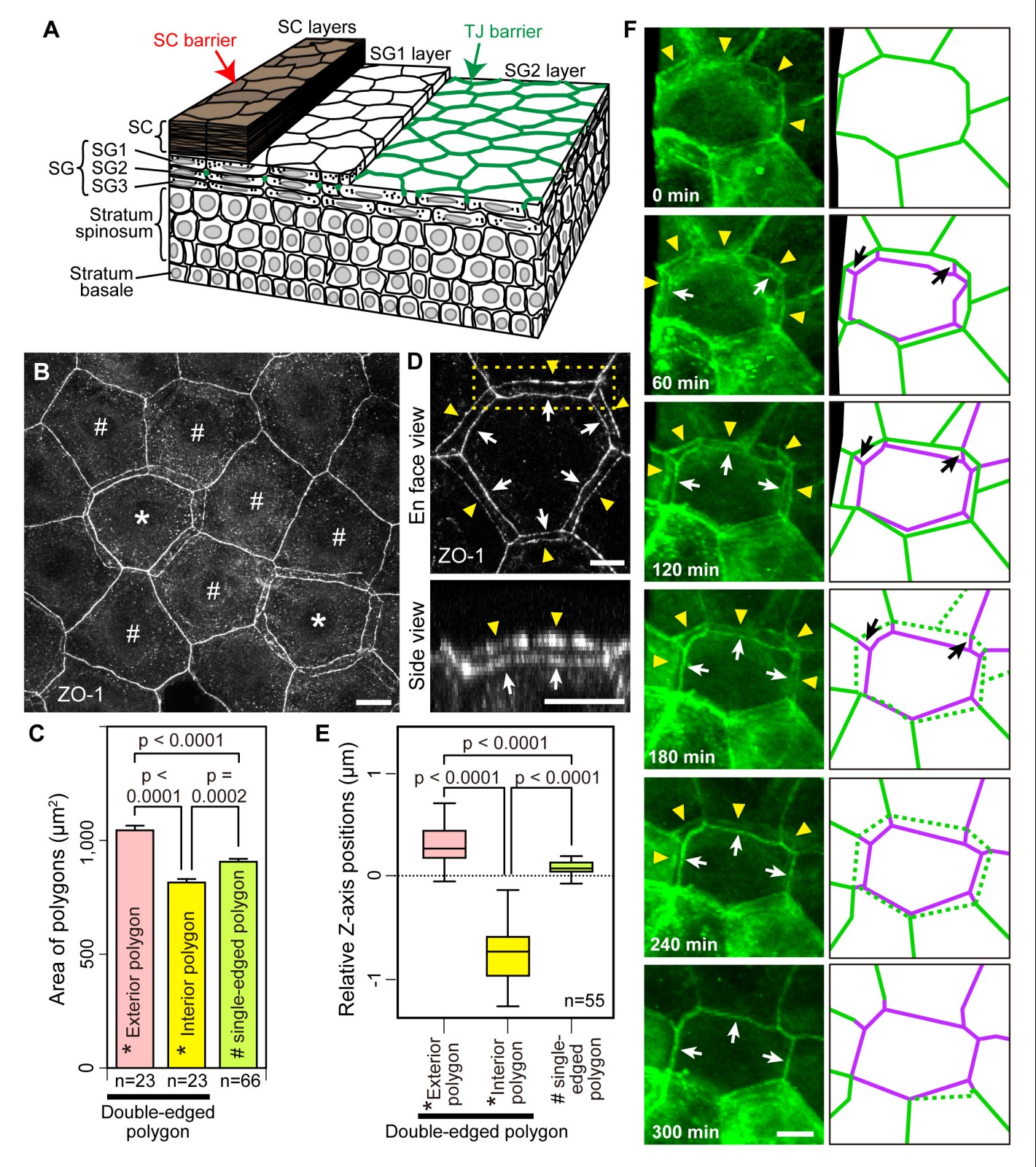

**Figure 1.** Multi-dimensional visualization of epidermal TJs and TJ-bearing cells in mouse-ear skin. (A) 3D structure of the epidermis. (B) *En face* image of ZO-1-positive honeycomb in mouse-ear epidermis showing double-edged polygons (*) and single-edged polygons (#). (C) Regularity in the size of the ZO-1-positive polygons represented in (B) and *Figure 1—figure supplement 1*, shown by the mean ± SEM [error bars] (one-way ANOVA multiple comparison test). (D) 3D image of a ZO-1-positive double-edged polygon in *en face* view (top) and 90°-rotated side view of the yellow-dotted rectangle

*Figure 1 continued on next page*

*Figure 1 continued*

(bottom). Upper exterior polygon, yellow arrowheads; lower interior polygon, white arrows. See *Video 1*. (E) Regularity of relative Z-axis position. Boxplots show the median, minimum, maximum, and interquartile range (one-way ANOVA multiple comparison test) for the ZO-1-positive polygons represented in *Figure 1—figure supplement 2*. (F) In vivo live images of Venus in the ear of ZO-1-Venus mice (left column) and their schematics (right column). Yellow arrowheads and green edges, edges of a Venus-positive polygon; white arrows and purple edges, edges of a newly appearing Venus-positive polygon; black arrows, Venus-positive edges connecting each vertex of the two polygons. See *Video 4*. Scale bars, 10 µm. TJ, tight junction; SC, stratum corneum.

The following source data and figure supplements are available for figure 1:

**Source data 1.** Percentage of double-edged polygons in ZO-1-positive honeycomb.
**Source data 2.** Size of the ZO-1-positive polygons.
**Source data 3.** Z-axis position of the ZO-1-positive polygons.
**Figure supplement 1.** Areas of exterior, interior and single-edged polygons.
**Figure supplement 2.** Relative Z-axis position of TJ polygons in TJ honeycomb evaluated in vivo.
**Figure supplement 3.** Epidermal TJ in ZO-1-Venus transgenic mice.

observations revealed a striking regularity in the size and relative Z-axis position of the inner and outer polygons compared to their adjacent single-edged polygons. The outer polygons were larger (1044.3 ± 20.6 µm$^2$, mean ± SEM, n = 23), and the inner ones smaller (814.7 ± 15.7 µm$^2$, n = 23), than their adjacent single-edged polygons (905.8 ± 12.6 µm$^2$, n = 66; *Figure 1C* and *Figure 1—figure supplement 1*). The inner (smaller) polygon was located lower in the epidermis than the outer (larger) one in each double-edged polygon (n = 99; *Figure 1D* and *Video 1*). Furthermore, comparison of the average Z-axis position of the vertices of the polygons revealed that the outer polygon was located significantly higher (0.294 ± 0.024 µm, n = 55), and the inner one significantly lower (−0.751 ± 0.036 µm, n = 55), than the adjacent single-edged polygons (−0.076 ± 0.009 µm, n = 55; *Figure 1E* and *Figure 1—figure supplement 2*). We designated the inner (smaller) and outer (larger) double-edged polygons as the interior and exterior polygons, respectively.

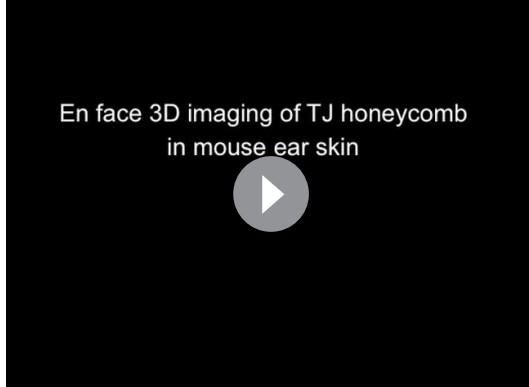

**Video 1.** *En face* 3D imaging of TJ honeycomb in mouse-ear skin. Representative 3D image of a ZO-1-positive double-edged polygon, shown in *Figure 1D*. TJ, tight jjunction.

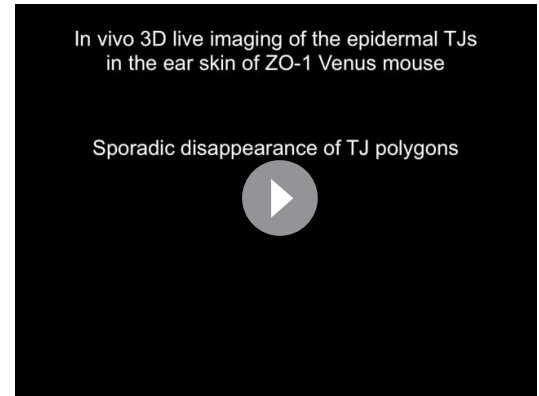

**Video 2.** In vivo 3D live imaging of epidermal TJs in the ear skin of a ZO-1 Venus mouse. Sporadic disappearance of TJ polygons (polygons marked with an asterisk). TJ, tight junction.

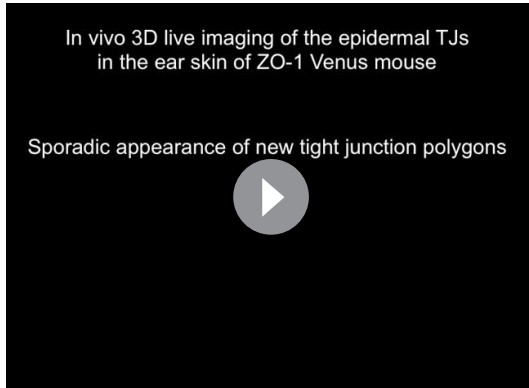

**Video 3.** In vivo 3D live imaging of epidermal TJs in the ear skin of a ZO-1 Venus mouse. Sporadic appearance of new TJ polygons (polygons marked with an asterisk). TJ, tight juction.

**Video 4.** In vivo 3D live imaging of edge-by-edge replacement of epidermal TJs in a particular polygon. Sequential appearance and disappearance of particular TJ polygons, shown in *Figure 1F*. TJ, tight junction.

## Double-edged TJ polygons appear sporadically during dynamic replacement of TJs

We next observed dynamic changes in the TJ honeycomb structure to identify regulatory mechanisms that maintain the TJ barrier. In vivo 3D live imaging of the transgenic mouse-ear skin, in which epidermal TJs were labeled with recombinant ZO-1 fused to the fluorescent protein Venus (*Nagai et al., 2002*) (*Figure 1—figure supplement 3*), revealed sporadic appearance and disappearance of TJ polygons (*Videos 2* and *3*). Closer observation of a particular TJ polygon identified the systematic temporal order of events for its dynamic appearance and disappearance. Namely, a new smaller (interior) polygon appeared beneath a pre-existing (exterior) polygon, forming a double-edged polygon as observed in the fixed samples (*Figure 1D*), followed by disappearance of the exterior polygon (*Figure 1F* and *Video 4*). Replacement of the old exterior polygon by a new interior polygon resulted in natural translocation of the cell body placed between the two polygons from the inside to the outside of the TJ barrier (discussed below in the cell turnover model, Figure 4B). These findings indicate that the double-edged polygons are where cells translocate across the TJ barrier and thus are the key structures for barrier homeostasis.

### Interior polygons show barrier function

Previous biotin permeation assays evaluated in vertical sections of mouse ear skin have not demonstrated barrier leakage at any TJ (*Furuse et al., 2002*; *Kubo et al., 2009*), suggesting that the barrier homeostasis of TJs is maintained even while the TJ polygons are being replaced. Does the interior polygon gain barrier function before its paired exterior polygon loses the barrier function during cell translocation across the TJ barrier?

We tested this hypothesis in whole mounted epidermis, using exfoliative toxin (ETA, MW = ~31 kDa [*Amagai et al., 2000*]) as a tracer that digests the extracellular portion of desmoglein 1 (Dsg1) (*Figure 2—figure supplement 1A–B*). The interior TJ polygons exhibited barrier function that protected the extracellular portion of Dsg1

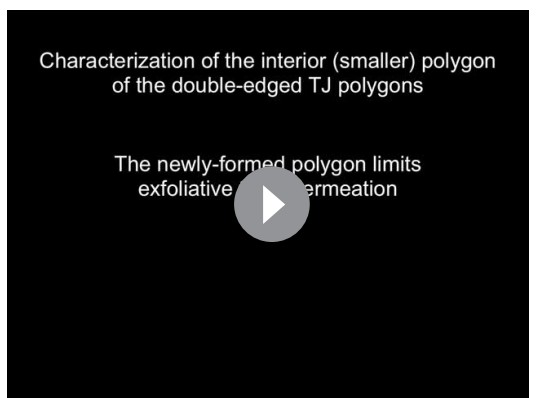

**Video 5.** Characterization of the interior (smaller) polygon of double-edged TJ polygons. The newly formed polygon limits ETA permeation (*Figure 2A–C*).

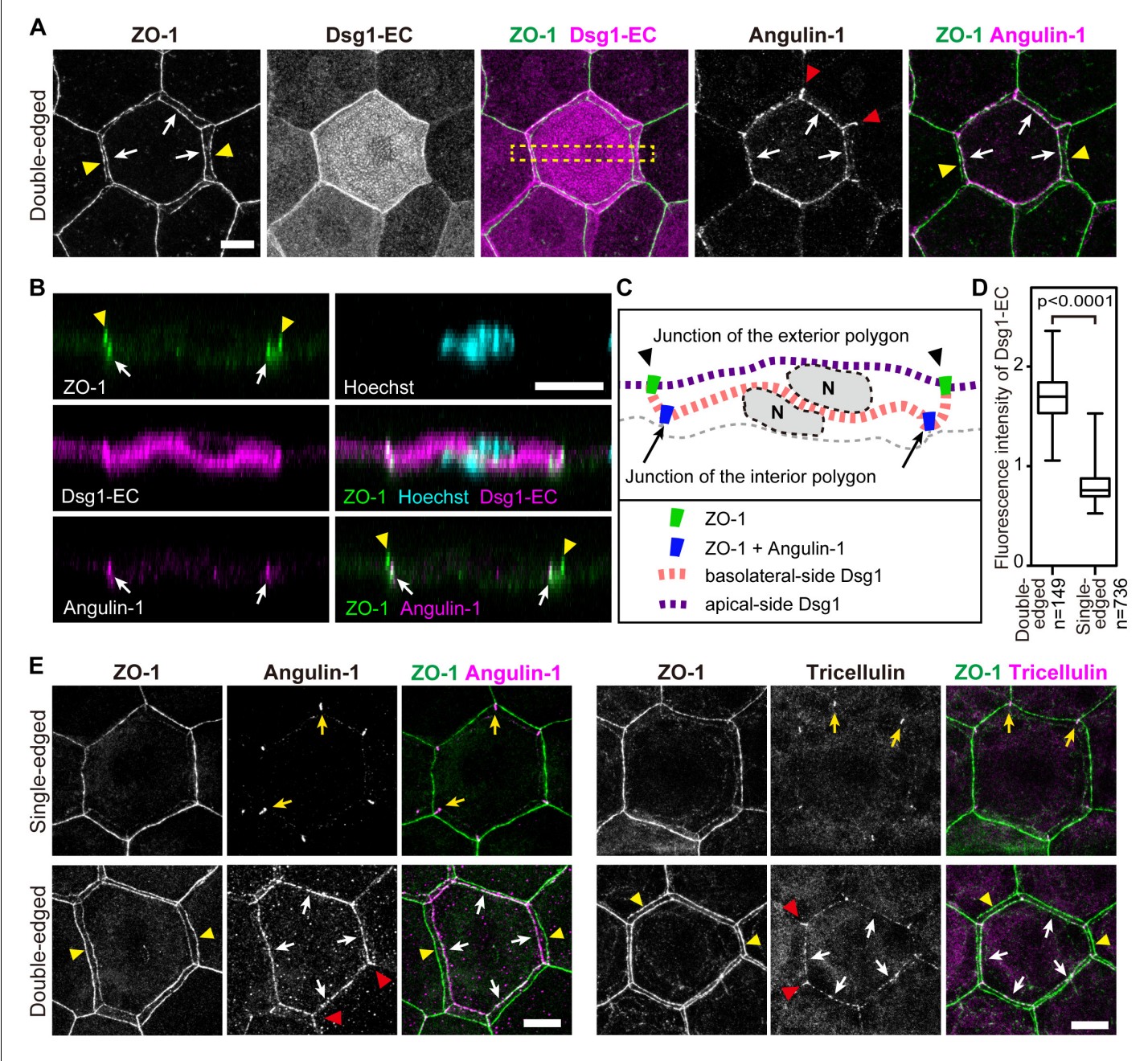

**Figure 2.** Characterization of interior polygons in the epidermal TJ honeycomb. (A and E) *En face* images of double-edged [A and lower panels in E] and single-edged [upper panels in E] TJ polygons in the ETA-induced cell sheet of the epidermis (*Figure 2—figure supplement 1*). (A) Subcellular localization of Dsg1-EC and angulin-1 at double-edged polygons. (B) 90°-rotated image of the yellow-dotted rectangle in (A) (see *Videos 5* and *6*). (C) Schematic of (B). N, nuclei; gray dotted lines, cell borders predicted from background cytosolic staining. (D) Fluorescence intensity of Dsg1-EC at the center of double- and single-edged polygons. The boxplots show the median, minimum, maximum, and interquartile range (Student's t-test). (E) Subcellular localization of tricellular TJ components (angulin-1 and tricellulin) at the single- and double-edged polygons. Yellow arrowheads, edges of the exterior polygon; white arrows, edges of the interior polygon; red arrowheads, vertical edges connecting the vertices of double-edged polygons; yellow arrows, vertices of single-edged polygons; Dsg1-EC, immunostained signals for extracellular portion of desmoglein 1, presumably representing desmosome-accumulating Dsg1. Scale bars, 10 μm.

The following source data and figure supplements are available for figure 2:

**Source data 1.** Fluorescent intensity of Dsg1-EC at single- and double-edged polygons.

*Figure 2 continued on next page*

Figure 2 continued

**Figure supplement 1.** Preparation of the exfoliative toxin (ETA)-induced cell sheet and isolation of TJ-bearing keratinocytes from the skin.

**Figure supplement 2.** Subcellular localization of claudin-1 at double-edged polygons.

against ETA (representative images, *Figure 2A–C* and *Video 5*; quantitative data of double-edged polygons [1.689 ± 0.020 times, mean ± SEM, n = 149] and their adjacent single-edged polygons [0.810 ± 0.006 times, n = 736], *Figure 2D*). These observations indicate that the development of the occlusive paracellular barrier of the interior TJ polygons prevents barrier leakage during cellular translocation from inside to outside the TJ barrier (discussed below in Figure 4B and C). Details of the barrier function of the interior TJ polygon, such as a dependency on molecular size or electric charge, remain to be determined, due to the limitations of in vivo permeation assays in stratified epithelia.

## Tricellular TJ components are localized on the interior polygon

To characterize the molecular players that mediate the formation of the double-edged polygons, we further investigated the newly formed interior polygons. Claudin-1, a major transmembrane protein of epidermal TJ (*Furuse et al., 2002*), was localized to both the exterior and interior polygons (*Figure 2—figure supplement 2*). In simple epithelia, most TJs are formed as conventional bicellular TJs (bTJs) on the apical edges between two cells, and tricellular TJs (tTJs) are formed at tricellular contacts among three cells (*Figure 4—figure supplement 1A and B*) (*Furuse et al., 2014*). In stratified epithelia of the epidermis, we observed the localization of tTJ-specific proteins (angulin-1 and tricellulin) on all edges of the interior polygons of the double-edged ZO-1-positive polygons, while the exterior polygons were bTJs (*Figure 2B,E* and *Video 6*). This is a striking contrast to the single-edged polygons, where tTJ-specific proteins were found as dots or short lines at the vertices as in simple epithelia (*Figure 2E*). These findings indicate that the interior TJ polygons consist of tTJs, suggesting that the edges of the interior polygons are tricellular contacts between three TJ-forming cells (discussed below in Figure 4B and *Figure 4—figure supplement 1*).

## The basic shape of SG2 cells is a flattened Kelvin's tetrakaidecahedron with TJs on its edges

Conventional schemas of skin often describe epidermal keratinocytes as rectangles in vertical 2D sections and assume their 3D cell shape to be a hexagonal prism (*Figure 3A*) (*Goldsmith et al., 2012*). Indeed, corneocytes are regularly stacked in the murine ear SC (*Mackenzie, 1969*; *Christophers, 1972*). However, the regular stack of hexagonal prism cells is inconsistent with our in vivo observations of the regularity of the sizes of TJ polygons (*Figure 1C* versus *Figure 3—figure supplement 1*). Moreover, the hexagonal prism structure would require unrealistic sliding of the cell columns for cell turnover, resulting in the breakage of the TJ barrier (*Figure 3—figure supplement 1*).

Our in vivo observations can be coherently explained if we assume the SG cells have the shape of the Kelvin's tetrakaidecahedron, which was originally proposed by Lord Kelvin in 1887 as the optimal natural space-filling shape with minimal surface area (*Figure 3B*) (*Thomson, 1887*). A flattened variation of Kelvin's tetrakaidecahedron (f-TKD) shape (*Figure 3C*) was once observed in the corneocytes at the surface of murine ear skin

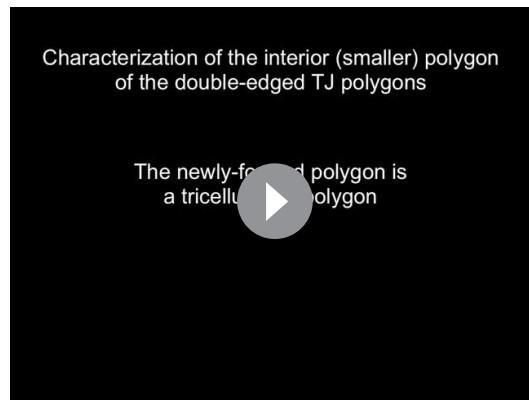

**Video 6.** Characterization of the interior (smaller) polygon of double-edged TJ polygons. The newly formed polygon is a tTJ polygon (*Figure 2E*).

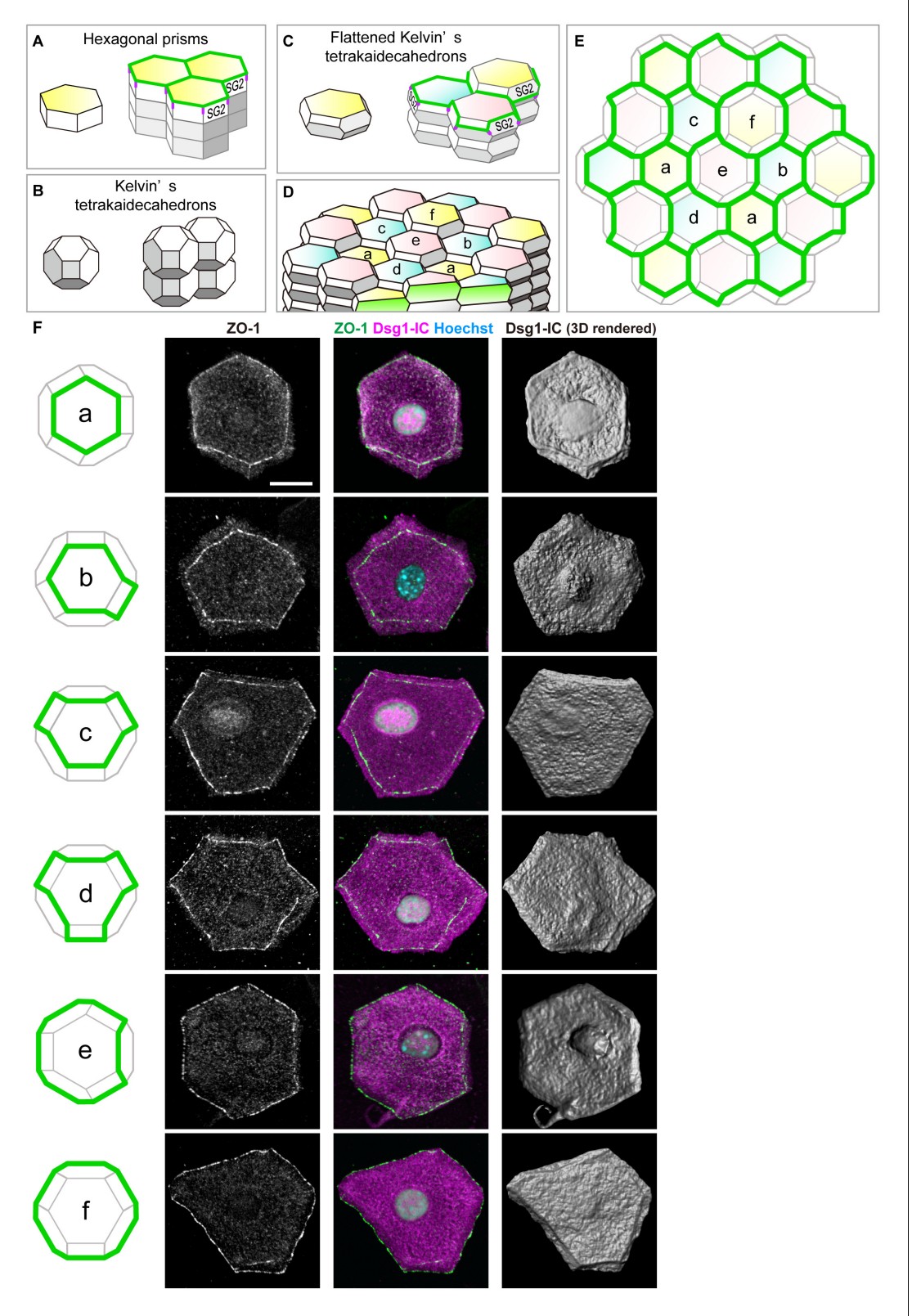

**Figure 3.** Characterization of the shape of SG2 cells and structures of epidermal TJ honeycomb. (A–C) Three possible space-filling structures of the SG. (D) Regular interdigitated stacks of f-TKD cells. SG2 cells are displayed at the top of cell columns. (E) *En face* view of the TJ honeycomb (green edges) on the f-TKD cell stacks shown in (D). (F) Six representative polyhedral shapes (a–f) of isolated SG2 cells, corresponding to the TJ polygons (green edges) in (D) and (E). The cells were visualized by cytoplasmic staining for the intracellular portion of desmoglein 1 (Dsg1-IC, middle column), with ZO-

*Figure 3 continued on next page*

*Figure 3 continued*

1-positive TJ at the edges (left and middle column), and their 3D rendered images (right column). See *Video 7* for 3D rendered images. TJ, tight junction.

The following figure supplement is available for figure 3:

**Figure supplement 1.** Possible cell turnover and TJ replacement in the conventionally proposed structure of SG and SC with hexagonal prism cells.

in scanning electron microscopic studies (*Allen and Potten, 1976*; *Menton, 1976*) but has been overlooked for several decades.

To investigate whether the 3D shape of SG cells is f-TKD, similarly to corneocytes, we isolated SG2 cells from mouse ear epidermis via sequential treatment with ETA and trypsin (*Figure 2—figure supplement 1A–C*). The immunofluorescent staining of isolated SG2 cells and their 3D rendered images demonstrated that the TJ-bearing SG2 cells have an f-TKD–like polyhedral shape, with TJs on their edges (*Figure 3F* and *Video 7*). Moreover, in vivo observations of the isolated SG2 cells (*Figure 3F*) confirmed the variations in the shapes of TJ polygons in the regular stacks of f-TKD cells, depending on their relative Z-axis position (*Figure 3D and E*). We thus concluded that the basic shape of SG2 cells is f-TKD.

## The flattened Kelvin's tetrakaidecahedron model

By integrating our in vivo observations on replacements of TJ polygons from edge to edge on the polyhedral cells, we propose an f-TKD turnover model, which describes a sophisticated spatiotemporal orchestration mechanism of cell-turnover to maintain the TJ barrier (*Figures 4* and *5*).

In this model, the cell turnover across the TJ barrier is completed in two phases (*Figure 4A and B* and *Video 8*). The cell turnover occurs at the cell column where there is an SG2 cell at a higher Z-axis position than any of its adjacent SG2 cells (phase 0). In the first phase (phase 1), an SG3 cell differentiates to become an SG2 cell, and then an interior TJ polygon is formed, leading to the appearance of a double-edged polygon, as observed in vivo (*Figure 1F*). The interior TJ polygon consists of tTJs formed among three SG2 cells: two vertically aligned SG2 cells and a laterally adjacent SG2 cell (*Figure 4B*). The upper SG2 cell naturally exits the TJ barrier, not by its own upward migration, but by the appearance of the interior TJ polygon on its basal side followed by the disappearance of the exterior TJ polygon. The precise subcellular structure of the interior TJ polygon predicted in the f-TKD model is shown in *Figure 4—figure supplement 1D*. The proposed model reproduces the in vivo observation of the regularity in the size of the TJ polygons (*Figure 1C* vs *Figure 4—figure supplement 2*) and in the Z-axis location of the polygons (*Figure 1E* vs *Figure 4A*).

In the second phase (phase 2, *Figure 4A and B*), the upper SG2 cell differentiates to become an SG1 cell, the exterior bTJ polygon disappears, and the interior tTJ polygon becomes a bTJ polygon. The precise order for formation of the functional TJs, whereby the interior TJ polygon gains barrier function before the exterior TJ polygon disappears (*Figure 4C*), is critical for TJ barrier homeostasis while cells are turned over.

After the second phase, the surrounding six f-TKD SG2 cells go through phases 0–2 and differentiate to become SG1 cells one by one (from

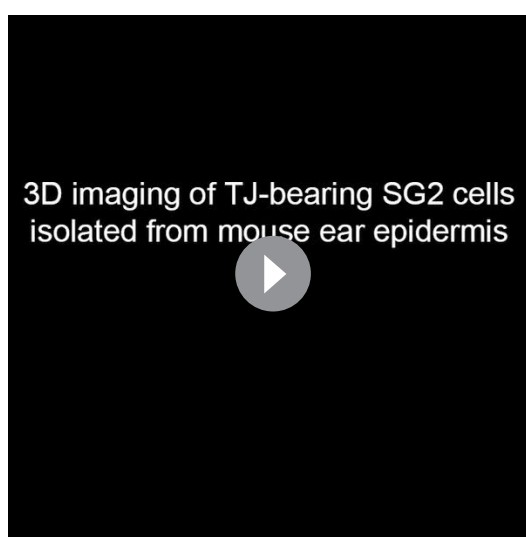

**Video 7.** Three-dimensional imaging of TJ-bearing SG2 cells isolated from mouse ear epidermis. Representative polyhedral shapes of isolated SG2 cells with TJs on their edges, shown in *Figure 3F*. TJ, tight junction.

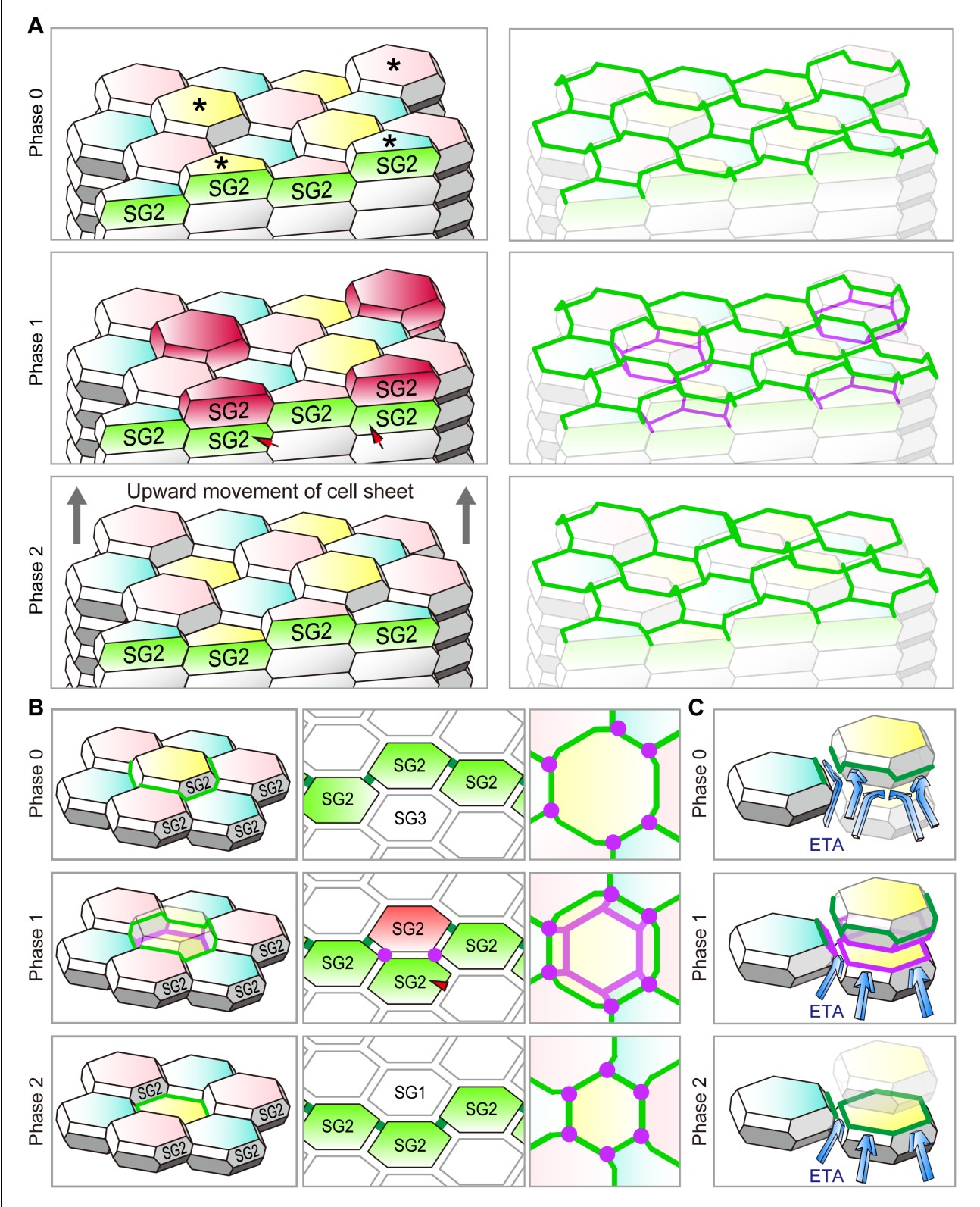

**Figure 4.** Cell turnover and TJ replacement in the SG2 f-TKD cells. (A and B) Cell turnover and TJ replacement in the SG2 layer of regular interdigitated stacks of f-TKD cells [3D structure of the cell stack with SG2 cells displayed at the top of cell columns, left panels in (A) and (B); 3D structure of the TJ barrier, right panels in (A); vertical sectional views, middle panels in (B); *en face* view of the TJ honeycomb, right panels in B]. Four SG2 cells (*) are located higher than their six adjacent SG2 cells in phase 0 in (A), and they differentiate to become SG1 cells in phase 2. The differentiation of SG3 cells

*Figure 4 continued on next page*

*Figure 4 continued*

to SG2 cells [red arrowheads in **A** and **B**] in phase 1 results in the vertical alignment of two SG2 cells, between which tTJ polygons (purple edges) are formed. Differentiation of the top SG2 cells (shown in red) to SG1 cells is completed in phase 2. (**C**) Schematics showing the occlusive function of single-edged TJ polygons (phase 0 and 2) and an interior tTJ polygon (phase 1) against the permeation of ETA. Green edges, bTJs; purple dots and edges, tTJs.

The following figure supplements are available for figure 4:

**Figure supplement 1.** Localization of TJ strands on the hexagonal prism cells in simple epithelia and on f-TKD cells in stratified epithelia.

**Figure supplement 2.** Regularity of polygon size in the TJ honeycomb on f-TKD cells in stratified epithelia.

#3 to #9; *Figure 5*), accompanied with an upward movement of the entire cell sheet (*Video 8*). As a result, the TJ polygon on the SG2 cell at the center increases in size in a stepwise manner via translocation of the cell from the lowest to the next highest of its six adjacent SG2 cells (from #3 to #9; *Figure 5*) until the next turnover cycle starts (from #9 to #11; *Figure 5*).

## Spatiotemporal regulation of dynamic TJ replacement maintains TJ barrier homeostasis despite continuous cell turnover

The characteristic feature of the proposed f-TKD turnover model (*Figure 5*) is that the spatiotemporal regulation of dynamic TJ replacement and continuous cell turnover across the TJ honeycomb can be explained by the following three simple local rules governing the f-TKD cells: (1) bTJs are formed between two SG2 cells, and tTJs are formed at the tricellular cell contact among three SG2 cells; (2) An SG2 cell located higher than its six adjacent SG2 cells differentiates to become an SG1 cell; (3) The differentiation of a SG3 cell to become a SG2 cell is synchronized with the differentiation of the pre-existing SG2 cell to become a SG1 cell in each cell column (*Figure 4B*).

To confirm that these three local rules ensure the maintenance of TJ barrier homeostasis, we developed a mathematical model implementing our f-TKD turnover model in silico. The in silico model consists of stacked layers of optimally packed SG cells, each of which is an f-TKD, similar to the geometric model of the corneocyte cornified envelope (*Feuchter et al., 2006*). The nominal values for the model parameters were determined so that the simulated dynamics reproduce the experimentally observed ratio of double-edged polygons among all the polygons (9.8 ± 0.6%, *Figure 1B*), as well as the average time required for cell turnover from entering to exiting the SG2 layer (24 hr), estimated from the average cell turnover rate in mouse ear epidermis (*Potten, 1975*). Computational simulation of the in silico model (*Figure 5—figure supplement 1*, and *Videos 9* and *10*) reproduced the entire sequence of TJ replacement on f-TKD cells, including formation of TJs, transition of the f-TKD cells, and upward movement of differentiated keratinocytes as a stratified cell sheet, while maintaining the relative location of individual f-TKD cells. It therefore confirmed the maintenance of TJ barrier homeostasis, cell-cell adhesion, and the physical strength of the cell sheet during cell turnover.

## Discussion

In this study, we demonstrated the biological links between the shape of TJ-forming cells and a mechanism for the maintenance of TJ barrier homeostasis in the epidermis, as a representative example of how tissues adopt form to follow function. Our proposed f-TKD cell turnover model suggests that the local spatiotemporal orchestration of cell differentiation in the SG cell layer enables the constituent f-TKD cells to be renewed while maintaining TJ barrier homeostasis in cornified epidermis.

The regular columnar stack of flattened corneocytes in murine ear epidermis demonstrates a regular zig-zag interdigitation pattern between two adjacent cell columns (*Figure 6—figure supplement 1* reproduced from Figure 1 of *Mackenzie [1975]*) (*Mackenzie, 1969*; *Christophers, 1972*; *Menton, 1976*; *Ball, 2001*). This regular interdigitation pattern was spontaneously reproduced by our f-TKD model (*Figure 6A*), in which cell differentiation in the SG2 cell layer occurs in turn in pairs

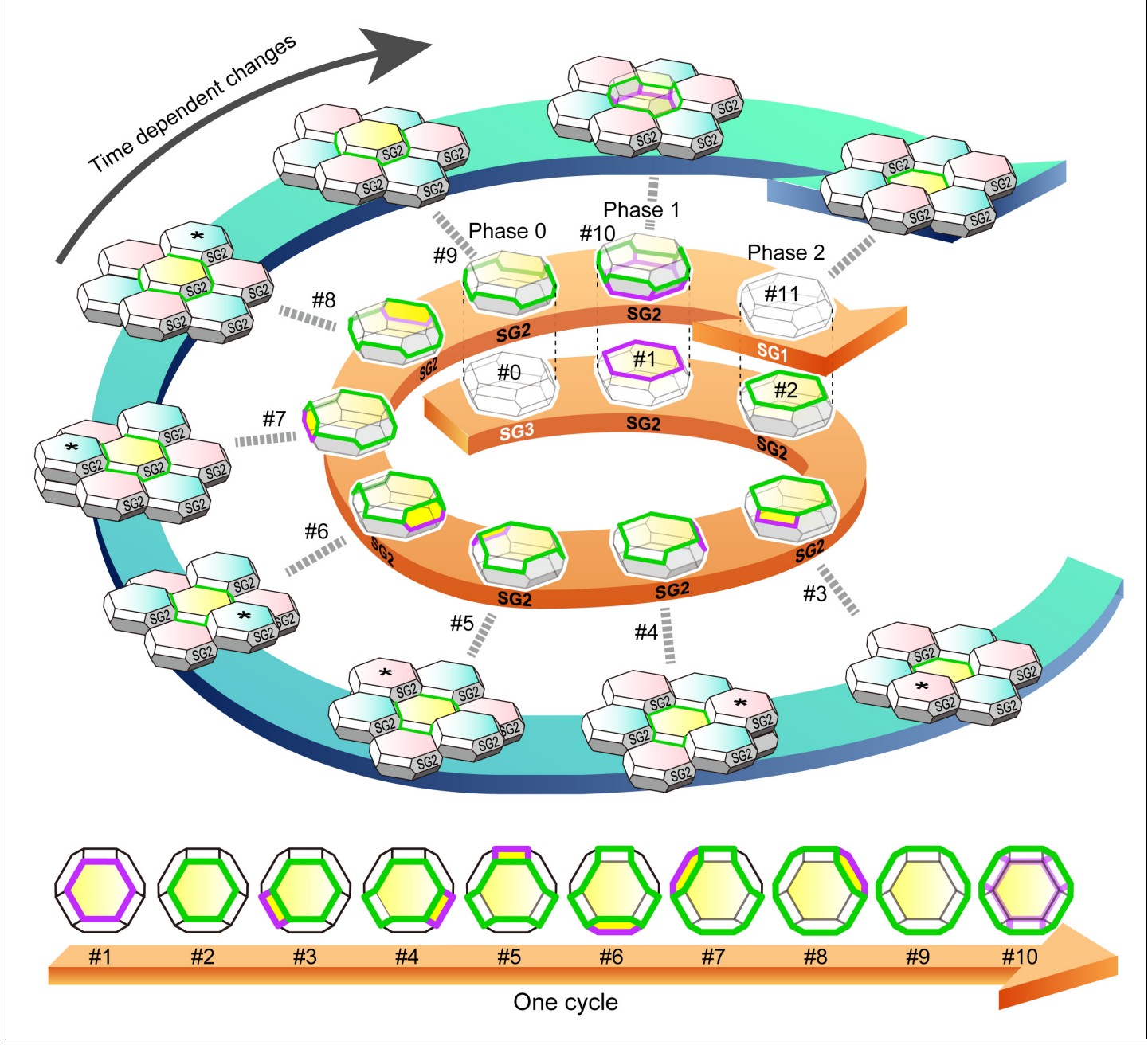

**Figure 5.** f-TKD turnover model. An entire cycle of the turnover of a set of f-TKD SG2 cells is shown on the outer blue spiral arrow, from the appearance of aTJ polygon (time point #1) to its disappearance (time point #11) on a particular SG2 f-TKD cell (the center yellow cell of the set of f-TKD cells). Three-dimensional cartoons and en face views of the yellow cell at each timepoint are depicted on the spiral and straight orange arrows, respectively. The relative Z-axis position of the cell changes from the lowest (time point #2) to the next highest (time point #9), while the adjacent SG2 cells (asterisked cells) are turned over. The TJ polygon of the cell becomes larger in a stepwise manner via edge-by-edge TJ replacement (timepoint #2–#9). The phases 0–2 correspond to those in *Figure 4B*. See *Video 8*. Green edges, bTJs; purple edges, tTJs.

The following figure supplement is available for figure 5:

**Figure supplement 1.** Computational simulation of f-TKD model for continuous cell turnover.

of adjacent cell columns. The f-TKD model thus provides a coherent mechanistic explanation of how the regular stacks of corneocytes are constructed.

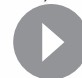

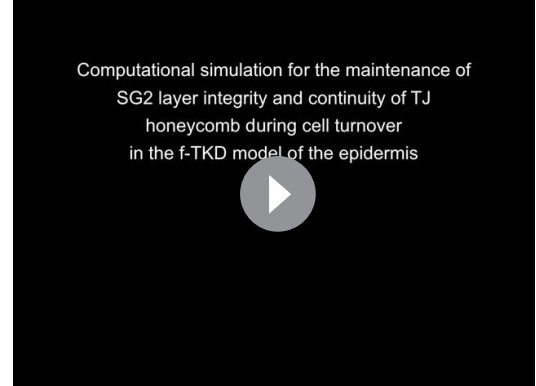

**Video 8.** Entire sequences of the edge-by-edge TJ replacement on a particular f-TKD cell (see *Figure 5*). DOI: 10.7554/eLife.19593.028

**Video 9.** 2D movie of computational simulation for the maintenance of SG2 layer integrity and continuity of TJ honeycomb during cell turnover in the f-TKD model of the epidermis. The 2D movie demonstrates the SG dynamics for 33 hr, with five frames per second and 7.2 min between frames. Green edges, conventional bTJ; purple edges, tTJ. bTJ (green edges) is formed at the apical edges of lateral SG2–SG2 cell contact faces, and tTJ (purple edges) is formed at the tricellular cell contact edges among three SG2 cells. tTJs at the vertices of single-edged TJ polygons were omitted from the display to better illustrate the honeycomb-mesh structure of TJ. Only the TJ-bearing SG2 cells are colored with three different colors depending on cell height. DOI: 10.7554/eLife.19593.029

The f-TKD model further suggests that homeostasis of the SC–SG layers is maintained by the spatiotemporal orchestration of cell differentiation in the SG2 cell layer, rather than by the kinetics of stem cell proliferation and differentiation in the basal layer. Our model is consistent with a pioneering computational simulation sug-

gesting that the regular 3D stacking structure of the SC can be spontaneously formed by randomly supplied cells (*Honda et al., 1996*). Our model also accords with in vivo cell-tracing studies demonstrating a random supply of cells from the spinous layer to the SG layer (*Doupé et al., 2010*), and in vivo live observations of the upward movement of spinous layer cells funneling into preexisting cell columns of the SG (*Rompolas et al., 2016*).

In the 1980s, the epidermal proliferative unit (EPU) concept postulated that each column of flattened corneocytes corresponds to a set of basal layer stem cells that proliferate directly under the column (*Potten and Allen, 1975*; *Potten, 2004*). In the EPU model, the regular interdigitation pattern of the SC is explained by the regular kinetics of basal layer stem cells in each EPU. However, recent in vivo cell-fate-tracing studies demonstrated more random cell fate decisions in the basal cell layer, with EPU model-like upward cell movement funneling into the cell columns of the SG, leading to a new concept: the epidermal differentiation unit (EDU) (*Rompolas et al., 2016*). However, the

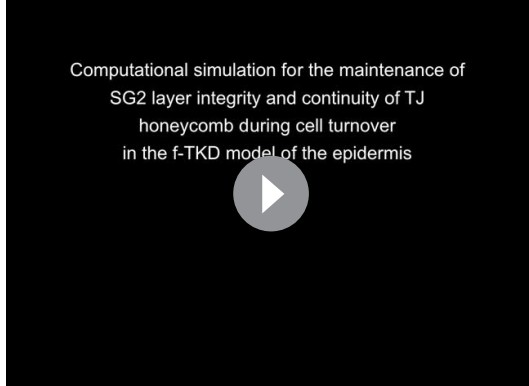

**Video 10.** 3D movie of computational simulation for the maintenance of SG2 layer integrity and continuity of TJ honeycomb during cell turnover in the f-TKD model of the epidermis. The 3D movie demonstrates the SG dynamics for 66 hr, with 20 frames per second and 7.2 min between frames. Green edges, conventional bTJ; purple edges, tTJ. bTJ (green edges) is formed at the apical edges of lateral SG2–SG2 cell contact faces, and tTJ (purple edges) is formed at the tricellular cell contact edges among three SG2 cells. tTJs at the vertices of single-edged TJ polygons were omitted from the display to better illustrate the honeycomb-mesh structure of TJ. Only the TJ-bearing SG2 cells are colored with three different colors depending on cell height. TJ, tight junction. DOI: 10.7554/eLife.19593.030

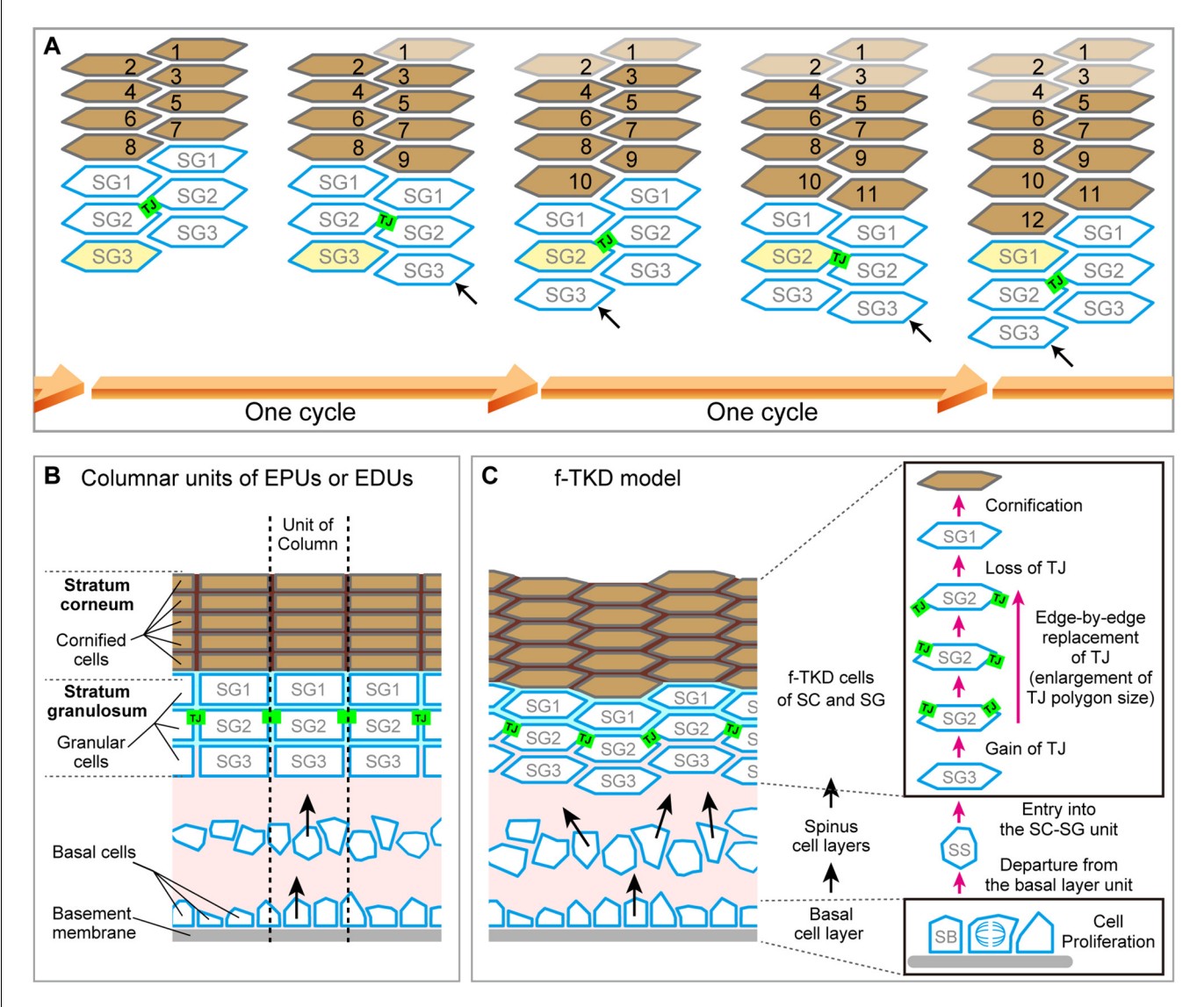

**Figure 6.** Spatiotemporal orchestration of cell differentiation in the f-TKD model generates interdigitated stacks of corneocytes. (A) The f-TKD model provides a coherent explanation of how the regular interdigitation of corneocytes is produced in the SG2 layer. Arrows show SG3 cells that are newly aligned to the columnar stack. One cycle of cell turnover (orange arrow) corresponds to one cycle indicated in *Figure 5*. The yellow-colored cell is differentiated from SG3 to SG1 in the time course. (B) Previously proposed columnar unit concept of epidermal structure and turnover. (C) Our proposed f-TKD model for epidermal homeostasis.

The following figure supplement is available for figure 6:

**Figure supplement 1.** Interdigitated stacks of corneocytes as a footprint of spatio-temporally regulated differentiation in SG2 cells.

underlying mechanism dictating the regular interdigitation pattern of cell columns remains enigmatic (*Figure 6B*).

In our f-TKD model, the cell columns exist only in the SG and SC layers, rather than extending from the basal layer through to the SC. The cells originate from stem cells in the basal layer and are randomly supplied to a spinous layer. Once the cells enter the SG layer, cell turnover in adjacent columns is tightly coordinated in a spatiotemporal manner, leading to the regular interdigitation pattern (*Figure 6C*). Epidermal homeostasis is maintained by balancing cell proliferation in the basal layer, cell translocation (differentiation) from the basal to the spinous layer, cell integration to the

SC/SG layer, and cell shedding from the top of the SC as squames. Future studies are needed to explore how this balance among critical processes in the epidermal layers is regulated to maintain a constant thickness of the epidermis.

The mammalian epidermis is a representative stratified epithelium. Nutrients for stratified cells are mostly supplied from the basal connective tissue via diffusion through paracellular pathways. If all the keratinocytes in the epidermis formed TJs, cells located in the upper epidermis would likely starve due to their segregation from the nutrient supply by multi-layered TJ barriers. Therefore, it is biologically reasonable that the TJ barrier is single-layered in stratified epithelia (*Kubo et al., 2012*; *Yoshida et al., 2013*). Our observations revealed that TJ formation is restricted only between SG2 cells (*Figure 4—figure supplement 1E*). The molecular mechanisms that coordinate the sequential differentiation steps from SG3 to SG1 cells and restrict TJ-forming activity to SG2 cells are currently unknown. The f-TKD model may help to reveal these mechanisms in future studies.

Various mammalian epidermis shows a regular interdigitation pattern in the SC (*Christophers, 1972*; *Mackenzie et al., 1981*), suggesting that the f-TKD cell turnover mechanism governs cell differentiation in mammals. Further investigation on whether this characteristic interdigitation pattern is observed in the SC of other vertebrates, such as amphibians, reptiles and birds, may reveal the general applicability of the f-TKD model to cornified stratified epithelia. Other mechanisms could be involved in maintaining TJ barrier homeostasis in simple epithelia and non-cornified stratified epithelia, where apoptotic cells are extruded to the outside TJ barrier by adjacent cells that migrate into the basal side of the apoptotic cells and form multi-junctional TJs (*Pentecost et al., 2006*; *Eisenhoffer and Rosenblatt, 2011*).

The actual structure of the optimal space-filling shape with minimal surface area could be more complex than Kelvin's model (*Lewis, 1943*; *Williams, 1968*; *Weaire and Phelan, 1994*), and the shape and alignment of corneocytes are much more variegated in human skin compared to mouse ear skin (*Mackenzie et al., 1981*). Nonetheless, the basic concept of the spatiotemporal orchestration of SG cell differentiation in the f-TKD model would be sufficient to explain the regular interdigitation pattern of corneocytes observed in various types of cornified skin (*Mackenzie et al., 1981*). The f-TKD cell turnover model of the SG can be applied to stratified stacks of variously shaped polyhedral cells and provides a fundamental basis for the maintenance of barrier homeostasis during cell turnover in cornified stratified epithelia.

## Materials and methods

### Animals

Female 8- to 12-week-old C57B6/J mice were used as wild type in all experiments. To establish ZO-1-Venus transgenic mice, a transgenic vector was constructed with the involucrin promoter vector (pH3700-pL2 (*Carroll and Taichman, 1992*), kindly provided by Dr. Lorne Taichman), which contained the first involucrin intron, an SV40 intron, a β-galactosidase gene, and an SV40 polyadenylation site. Mouse cDNA encoding the full length of ZO-1 and Venus (*Nagai et al., 2002*) cDNA (kindly provided by Dr. Atsushi Miyawaki) was replaced with the β-galactosidase gene in the involucrin promoter vector. The transgenic vector was injected into the fertilized egg from C57B6/J male and F1 female of C57B6/J × C3H. Transgenic mice were screened by direct observation of mouse-ear skin using fluorescence microscopy and the involucrin-promoter-driven Venus-ZO-1 mouse line was established by crossing with Balb/c mice more than eight times (RRID:MGI:5805289). All animal protocols were approved by the Animal Ethics Review Board of Keio University and conformed to the National Institutes of Health guidelines.

### Immunofluorescence and confocal microscopy

Mouse skin samples were embedded in optimal cutting temperature compound (Sakura Finetek, Japan), frozen in liquid nitrogen, and sectioned using a cryostat, as described previously (*Yoshida et al., 2013*). The frozen sections were processed immediately by incubation in 95% ethanol at 4°C for 30 min, followed by 100% acetone at room temperature for 1 min and immunostained as described previously (*Yokouchi et al., 2015*). Epidermal sheets were prepared from the ventral side of mouse-ear skin and immunostained, as described previously (*Yokouchi et al., 2015*). For the preparation of ETA-induced SC-SG sheets (*Figure 2—figure supplement 1*), 100 μL of 44 μg/mL

recombinant ETA (*Hanakawa et al., 2002*) in PBS containing 1 mM CaCl$_2$ was injected intradermally into the ventral skin of the mouse ear and incubated for 30 min at 37°C. The ETA-induced SC-SG sheets were stripped away from the ventral skin, fixed by incubation in 95% ethanol on ice for 30 min, and immunostained, as described previously (*Yokouchi et al., 2015*). Cell sheet samples were mounted in a whole-mount fashion using Mowiol (Millipore, Germany). Samples were observed under a Leica TCS sp5 laser scanning confocal microscope equipped with a 63× objective using 0.4–0.5 μm optical slices. 3D reconstruction images were built using Leica sp5 software and Imaris software (Bitplane, Switzerland ). Images and movies were processed using Adobe Photoshop CS6, Adobe Illustrator CS6, and Apple QuickTime Pro.

## Antibodies

The following primary antibodies were used: polyclonal antibodies against claudin-1 (ab15098; Invitrogen, Carlsbad, CA, RRID:AB_301644), intracellular portion of desmoglein1 (sc20114; Santa Cruz Biotechnology, Dallas, TX, RRID:AB_2293011), angulin-1 (lipolysis-stimulated lipoprotein receptor) (*Masuda et al., 2011*) at a 1:200 dilution, monoclonal antibodies against ZO-1 (T8-754; kindly provided by Dr. Masahiko Itoh) (*Itoh, 1991*) at a 1:10 dilution, tricellulin (kindly provided by Dr. Sachiko Tsukita) (*Ikenouchi et al., 2005*), and occludin (MOC37) (*Saitou et al., 1997*). The extracellular portion of desmoglein one was detected by single-chained scFv (3–30/3 hr) (*Ishii et al., 2008*; *Yoshida et al., 2013*). Species-specific secondary antibodies and streptavidin-labeled Alexa Fluor 488, 568, and 647 (Invitrogen) were used for detection at a 1:200 dilution. Cell nuclei were stained with Hoechst 33258 (Invitrogen).

## Permeation assay

A TJ permeation assay was performed according to a modified version of the procedure described previously (*Yokouchi et al., 2015*). For the permeation assay with a protein biotinylation reagent (*Figure 2—figure supplement 1*), 30 μL of 10 μg/mL recombinant ETA (*Hanakawa et al., 2002*) in PBS containing 1 mM CaCl$_2$ was injected into the dermis on the ventral side of the mouse ear, followed by injection of 50 μL of 10 mg/mL Sulfo-NHS-LC-Biotin (556 Da, #21335; Thermo Fisher Scientific, Waltham, MA) 10 min later. After a 30-min incubation, skin samples were biopsied and embedded in optimal cutting temperature compound (Sakura Finetek). For the ETA permeation assay, 50 μL of 10 μg/mL recombinant ETA in PBS containing 1 mM CaCl$_2$ was injected intradermally on the ventral side of adult mouse ears (*Hanakawa et al., 2002*; *Yoshida et al., 2013*). After 30 min, the skin was biopsied and the ETA-induced bulla roof was stripped away and fixed by incubation in 95% ethanol on ice for 30 min and immunostained.

The barrier function of the interior polygons against ETA permeation was quantitatively analyzed as follows. ETA-induced bulla roof was immunostained for ZO-1, the extracellular portion of Dsg1, angulin-1 and nuclei in five independent studies. Using a confocal microscope, 20 square *en face* images (15,376 μm$^2$) were taken from each bulla roof of five mice. To quantitatively compare the single- and double-edged polygons, the relative fluorescent intensity of the extracellular portion of Dsg1 at each pixel was determined in the *en face* image by calculating the ratio of its fluorescence against the total average fluorescent intensity of the image. The fluorescent intensity of each polygon was determined by its average at the center area (a circle of 320 μm$^2$).

## Isolation of SG2 cells

ETA-induced SC–SG cell sheets were prepared as described above. The cell sheets were floated on 500 μL of a 1:1 mixed solution of 0.5 g/L trypsin/0.53 mmol/L EDTA solution (32778–34, Nacalai Tesque, Japan) and 2.5 g/L trypsin solution (35555–54, Nacalai Tesque) and incubated at 37°C for 20 min. Cells were dissociated from the SC by pipetting. The trypsin solution containing SG cells was diluted with 1 mL of PBS, passed through a cell strainer (100 μm diameter pores, 352360, Corning, Corning, NY), collected in a 15-mL conical tube (Corning), and centrifuged at 180 G for 5 min at room temperature. The pellet was fixed in 10 mL of 95% ethanol on ice for 30 min. The fixed cells were pelleted by centrifugation at 720 G at room temperature for 5 min, dissociated in 1 mL of PBS, collected on a slide glass using cytospin (Thermo Fisher Scientific) at 450 G at room temperature for 5 min, and immunostained as described above. The shape of the isolated SG2 cells was visualized by immunostaining for the cytoplasmic portion of Dsg1 (i.e., staining for the precursor

proteins in the endoplasmic reticulum and cytoplasmic vesicles and for the trypsin-digested proteins in the remaining desmosomes and endosomes). Three-dimensional rendering of the immunostained SG2 cells was performed by Imaris software (Bitplane).

## In vivo 3D live imaging of mouse epidermal TJs using two-photon laser scanning microscopy

ZO-1 Venus mice were anesthetized with isoflurane (AbbVie, North Chicago, IL) and oxygen and air; the isoflurane concentration was initially set at 4% and gradually lowered to 1.2% in a constant oxygen flow (0.15 L/min). To prevent movement, the dorsal side of the mouse ear was fixed to the table with double-sided adhesive tape. The two-photon microscope is a custom-made upright microscope (BX61WI, Olympus, Japan) attached to a mode-locked titanium-sapphire laser system (Chameleon Vision II, Coherent, Santa Clara, CA) that achieves a 950 nm laser with a 140-fs pulse width and an 80-MHz repetition rate (*Morikawa et al., 2012*). Images (512 × 512 pixels) were acquired by z-stack scanning at 0.6 μm intervals with a 25× objective lens (XLPLN25 × WMP; NA 1.05, Olympus) and an Olympus FV1000 scanning unit using Fluoview software (FV10-ASW, Olympus). Emitted fluorescence was detected using an external photomultiplier tube (R3896; Hamamatsu Photonics, Japan) after reflection *via* a dichroic mirror (580 nm cut-off) and passing through an emission filter (500–550 nm). Acquired images were processed using Imaris software (Bitplane).

## Counting and analyzing the double-edged polygons

Epidermal sheets of mouse-ear skin were immunostained for ZO-1 in five independent assays as described above. Using the confocal microscope, 20 square *en face* images of 15,376 μm$^2$ were taken from each sample. The number of single- and double-edged polygons was counted manually. The percentage of double-edged polygons was determined as the average of five independent assays.

The size of single- and double-edged polygons (*Figure 1—figure supplement 1*) was measured using ImageJ software (NIH), for 10 square *en face* images (15,376 μm$^2$) that included 8–10 TJ polygons with their whole edges. Comparison of the Z-axis relative position between polygons was performed on the double-edged polygons surrounded by six single-edged polygons for 55 square *en face* images (15,376 μm$^2$). The Z-axis position of a polygon was defined as the average of the Z-axis positions of its vertices analyzed by Imaris software (*Figure 1—figure supplement 2*). The relative Z-axis position of a polygon (*Figure 1E*) was calculated with respect to the average of the Z-axis position for a set of eight polygons (an exterior and an interior polygon, and their six adjacent single-edged polygons).

## Statistical analyses

We used Prism six software (GraphPad, La Jolla, CA) for all statistical analyses.

## Computational simulation of f-TKD turnover model

We developed a 3D in silico f-TKD turnover model consisting of stacked layers of optimally packed SG cells, each of which was an f-TKD. TJ-bearing SG2 cells were shown in three different colors depending on the vertical placement of the cell. A cycle of the SG2-to-SG1 transition of a cell starts with the appearance of the tTJs and is completed by their disappearance. In our model simulation, the SG2-to-SG1 transition of a cell is demonstrated by the gradual change of an SG2 cell to a transparent SG1 cell. The transition of a cell is governed by two local rules: (1) The differentiation of a SG3 cell to become a SG2 cell is synchronized with the differentiation of the pre-existing SG2 cell to become a SG1 cell in each cell column. (2) An SG2 cell located higher than its six adjacent SG2 cells differentiates to become an SG1 cell after a waiting period that is stochastically determined with a uniform probability between 0 and 9.6 hr. This uniform distribution was chosen to best reproduce the average turnover time (24 hr), estimated from the experimentally observed cell turnover rate in mouse ear skin (*Potten, 1975*) and the percentage of double-edged polygons (9.8%, *Figure 1B*). The average time required to complete the TJ disappearance during SG2-to-SG1 transition was set to be 2.4 hr = 24 × 9.8/100.

Additionally, the following two rules for TJ production were implemented in the algorithm: (1) Bicellular TJs (green) are produced on edges shared by two SG2 cells when edges are in contact

with SG1 cells; (2) tTJs (blue) are produced on edges shared by three SG2 cells. In our simulations, tTJs on double-edged polygons (where TJ replacement takes place) are displayed, but tTJs at vertices of single-edged polygons are not displayed for the sake of simplicity (*Figure 5—figure supplement 1*, *Videos 9* and *10*). All simulations were conducted using Matlab version R2015b (The MathWorks, Natick, MA).

## Acknowledgements

We are grateful to J R Stanley for critical reading of the manuscript, M Okajima for laboratory management, and K Ishii, H Ito and S Sato for technical assistance. This work was supported in part by Grants-in-Aid for Scientific Research from the Ministry of Education, Culture, Sports, Science and Technology of Japan; a Health Labour Sciences Research Grant for Research on Allergic Diseases and Immunology from the Ministry of Health, Labour and Welfare of Japan; Japan Agency for Medical Research and Development (AMED); Maruho Co., Ltd; The Mochida Memorial Foundation for Medical and Pharmaceutical Research; and Lydia O'Leary Memorial Pias Dermatological Foundation. This manuscript is dedicated to the memory of Dr. Shoichiro Tsukita.

## Additional information

### Funding

| Funder | Author |
| --- | --- |
| Ministry of Health, Labour and Welfare | Masayuki Amagai |
| Japan Agency for Medical Research and Development | Masayuki Amagai |
| Ministry of Education, Culture, Sports, Science, and Technology | Akiharu Kubo |
| Lydia O'Leary Memorial Pias Dermatological Foundation | Akiharu Kubo |
| Mochida Memorial Foundation for Medical and Pharmaceutical Research | Akiharu Kubo |
| Maruho Co., Ltd | Masayuki Amagai |

The funders had no role in study design, data collection and interpretation, or the decision to submit the work for publication.

### Author contributions

MY, AK, Conception and design, Acquisition of data, Analysis and interpretation of data, Drafting or revising the article, Contributed unpublished essential data or reagents; TA, MK, MS, Acquisition of data, Analysis and interpretation of data, Contributed unpublished essential data or reagents; MvL, Analysis and interpretation of data, Drafting or revising the article, Contributed unpublished essential data or reagents; RJT, MA, Conception and design, Analysis and interpretation of data, Drafting or revising the article, Contributed unpublished essential data or reagents; MF, Conception and design, Drafting or revising the article, Contributed unpublished essential data or reagents

### Author ORCIDs

Reiko J Tanaka, http://orcid.org/0000-0002-0769-9382
Mayumi Kajimura, http://orcid.org/0000-0003-3773-6874
Makoto Suematsu, http://orcid.org/0000-0002-7165-6336
Mikio Furuse, http://orcid.org/0000-0003-2847-8156
Masayuki Amagai, http://orcid.org/0000-0003-3314-7052
Akiharu Kubo, http://orcid.org/0000-0003-0902-3586

## Ethics

Animal experimentation: All animal protocols were approved by the Animal Ethics Review Board of Keio University (Permit Number: 08070) and conformed to the Guide for the Care and Use of Laboratory Animals of the National Institutes of Health.

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
