## [Decision Letter]

Thank you for submitting your article "Epidermal cell turnover across tight junctions based on Kelvin's tetrakaidecahedron cell shape" for consideration by *eLife*. Your article has been favorably evaluated by Fiona Watt as the Senior Editor and three reviewers, one of whom is a member of our Board of Reviewing Editors.

The reviewers have discussed the reviews with one another and the Reviewing Editor has drafted this decision to help you prepare a revised submission.

General assessment:

Yokouchi and colleagues employs state of the art imaging technologies to reveal novel information regarding the structure and turnover of epidermal cells with tight junctions in the skin. The authors show that tight junctions appear with upper and lower polygons in the stratum granulosum (SG). Using observations on TJ shape and function, the present manuscript adapts Kelvin's tetrakaidecahedron to provide a very original, attractive and interesting mathematical and physical model that explains how the continuously renewing epidermal barrier can retain TJ barrier function in a single layer, the SG2, while individual cells move upward. Overall, this manuscript changes the paradigm for our understanding of how stratified cells with tight junctions continually regeneration and yet maintain barrier function. The work is logically presented and the data are reasonably thorough and convincing in support of these impactful conclusions.

Essential revisions:

The reviewers raise a number of concerns that must be adequately addressed before the paper can be accepted. Some of the required revisions will likely require further experimentation within the framework of the presented studies and techniques.

1) Direct evidence to support the f-TKD shape of the SG2 cells using 3D rendering is necessary. The authors could use a general cell surface membrane label (this can even be done after isolation of SG2 and SG1 cells using the ETA approach) or use phalloidin to label the cell cortex.

2) The model proposes that the structure identified is essential for epidermal barrier formation. Barrier function of this second TJ polygon is only shown by imaging for one cell in Figure 2. More extensive quantitative analysis, and perhaps an additional test such as biotin penetration, as the observation that the second ZO-1 positive polygon guarantees barrier function during transition is one of the key points of the paper and a corner stone of the model.

3) If possible, it would be useful to understand how generalizable these observations made in mouse ear are to other stratified epithelia and perhaps to non-keratinizing mucosal surfaces where a lipid-enriched stratum corneum is not present to help maintain barrier function.

4) The paper would be tremendously strengthened if the authors could quantitatively link the occurrence and disappearance of TJ polygons to the position of cells in relation to its neighbors.

---

## [Author Response]

*[…] Essential revisions:*

*The reviewers raise a number of concerns that must be adequately addressed before the paper can be accepted. Some of the required revisions will likely require further experimentation within the framework of the presented studies and techniques.*

*1) Direct evidence to support the f-TKD shape of the SG2 cells using 3D rendering is necessary. The authors could use a general cell surface membrane label (this can even be done after isolation of SG2 and SG1 cells using the ETA approach) or use phalloidin to label the cell cortex.*

In response to this comment, we performed 3D rendering of isolated SG2 cells, and the resulting direct evidence supporting the f-TKD shape of the SG2 cells is shown in the revised Figure 3 (Results, subsection “The basic shape of SG2 cells is a flattened Kelvin’s tetrakaidecahedron with TJs on its edges”, last paragraph and Materials and methods, subsection “Isolation of SG2 cells”).

We identified the polyhedral shape of the isolated SG2 cells and the TJs on their edges by staining for the intracellular portion of Dsg1 (anti-Dsg1 antibody, sc20114, Santa Cruz Biotechnology, Santa Cruz, CA, USA) and ZO-1. The observation of isolated SG2 cells confirmed variations in the shapes of the TJ polygons (Figure 3), as expected from the regular stacks of f-TKD cells (Figure 3).

*2) The model proposes that the structure identified is essential for epidermal barrier formation. Barrier function of this second TJ polygon is only shown by imaging for one cell in Figure 2. More extensive quantitative analysis, and perhaps an additional test such as biotin penetration, as the observation that the second ZO-1 positive polygon guarantees barrier function during transition is one of the key points of the paper and a corner stone of the model.*

In response to this comment, we quantitatively analyzed the barrier function of the interior ZO-1-positive polygons against ETA permeation. The results demonstrating the barrier function of the second TJ polygon were added in Figure 2, and in the text (Results, subsection “Interior polygons show barrier function” and Materials and methods, subsection “Permeation assay”, last paragraph).

We analyzed ETA permeation into stratified epithelia via en faceimaging because the ETA digestion of Dsg1 inside the TJ barrier results in total disappearance of the signals from the extracellular portion of Dsg1 at the inside TJ barrier (Figure 2—figure supplement 1). Other tracers, such as the biotin tracers suggested by the reviewer, are not appropriate, because the stratified intercellular staining in stratified epithelia hampers detailed analyses evaluating tracer permeation via en faceobservation. We delineated these potential difficulties in the Results (subsection “Interior polygons show barrier function”, last paragraph) to improve understanding.

*3) If possible, it would be useful to understand how generalizable these observations made in mouse ear are to other stratified epithelia and perhaps to non-keratinizing mucosal surfaces where a lipid-enriched stratum corneum is not present to help maintain barrier function.*

We added a discussion of simple epithelia and non-cornified stratified epithelia, in which another strategy for barrier maintenance has been observed (Discussion, seventh paragraph). We also expanded on the potential general applicability of the f-TKD model in the aforementioned paragraph.

*4) The paper would be tremendously strengthened if the authors could quantitatively link the occurrence and disappearance of TJ polygons to the position of cells in relation to its neighbors.*

In response to this comment, we performed statistical analysis on the Z-axis position of each polygon, which revealed remarkable regularity in that the exterior polygon was located significantly higher than, and the interior polygon was located significantly lower than, the adjacent single-edged polygons (Figure 1, Results, subsection “Double-edged TJ polygons are observed in the single-layered epidermal TJ honeycomb”, last paragraph and in Materials and methods, subsection “Counting and analyzing the double-edged polygons”, last paragraph). These observations support the f-TKD model, in which the occurrence and disappearance of TJ polygons are linked to the positions of cells relative to their neighbors.